# Anthropic Settlements' Impact on the Light-Absorbing Aerosol Concentrations and Heating Rate in the Arctic

Niccolò Losi [1,*], Piotr Markuszewski [2,3,4], Martin Rigler [5], Asta Gregorič [5,6], Griša Močnik [6,7], Violetta Drozdowska [2], Przemysław Makuch [2], Tymon Zielinski [2], Paulina Pakszys [2], Małgorzata Kitowska [2], Amedeo Manuel Cefalì [1], Irene Gini [1], Andrea Doldi [1], Sofia Cerri [1,8], Pietro Maroni [1], Ezio Bolzacchini [1] and Luca Ferrero [1]

1   GEMMA and POLARIS Centre, Università degli Studi di Milano-Bicocca, 20126 Milano, Italy; a.cefali@campus.unimib.it (A.M.C.); a.doldi1@campus.unimib.it (A.D.); s.cerri@campus.unimib.it (S.C.); p.maroni1@campus.unimib.it (P.M.); ezio.bolzacchini@unimib.it (E.B.); luca.ferrero@unimib.it (L.F.)
2   Institute of Oceanology, Polish Academy of Sciences, 81-712 Sopot, Poland; pmarkusz@iopan.pl (P.M.); drozd@iopan.pl (V.D.); makuch@iopan.pl (P.M.); tymon@iopan.pl (T.Z.); pakszys@iopan.pl (P.P.); gosiak@iopan.pl (M.K.)
3   Department of Environmental Science, Stockholm University, 10691 Stockholm, Sweden
4   Bolin Centre for Climate Research, Stockholm University, 10691 Stockholm, Sweden
5   Aerosol d.o.o., Kamniška 39A, SI-1000 Ljubljana, Slovenia; mrigler@aerosolmageesci.com (M.R.); agregoric@aerosolmageesci.com (A.G.)
6   Center for Atmospheric Research, University of Nova Gorica, SI-5000 Nova Gorica, Slovenia; grisa.mocnik@ung.si
7   Condensed Matter Physics Department, J. Stefan Institute, 1000 Ljubljana, Slovenia
8   Department of Environmental Sciences, Computer Science and Statistics, University of Ca' Foscari, 30172 Venezia Mestre, Italy
*   Correspondence: n.losi@campus.unimib.it

**Abstract:** Light-absorbing aerosols (LAA) impact the atmosphere by heating it. Their effect in the Arctic was investigated during two summer Arctic oceanographic campaigns (2018 and 2019) around the Svalbard Archipelago in order to unravel the differences between the Arctic background and the local anthropic settlements. Therefore, the LAA heating rate (HR) was experimentally determined. Both the chemical composition and high-resolution measurements highlighted substantial differences between the Arctic Ocean background (average eBC concentration of $11.7 \pm 0.1$ ng/m$^3$) and the human settlements, among which the most impacting appeared to be Tromsø and Isfjorden (mean eBC of $99.4 \pm 3.1$ ng/m$^3$). Consequently, the HR in Isfjorden ($8.2 \times 10^{-3} \pm 0.3 \times 10^{-3}$ K/day) was one order of magnitude higher than in the pristine background conditions ($0.8 \times 10^{-3} \pm 0.9 \times 10^{-5}$ K/day). Therefore, we conclude that the direct climate impact of local LAA sources on the Arctic atmosphere is not negligible and may rise in the future due to ice retreat and enhanced marine traffic.

**Keywords:** black carbon; Arctic; heating rate; direct radiative forcing

## 1. Introduction

The Arctic region has been experiencing a greater warming process than the rest of the globe since the 1980s, a process known as Arctic Amplification (AA) [1–6]. It has been demonstrated that AA has been increasing from ~1980 till the present after a period of Arctic cooling [7]; from ~1980 the Arctic temperature increase was two to four times faster than the rest of the globe, depending on the Arctic region [1,6] and season [8]. As a result, AA leads to a rapid reduction in sea-ice extent, earlier spring melt and a lengthening of the melt season, and land-ice retreat [2,9]. These changes are strengthened by local feedbacks, which in turn contribute to the AA, such as snow/ice albedo feedback and other feedbacks related to changes in heat fluxes between the ocean and the atmosphere and

changes in cloud cover and relative humidity, thus leading to longwave radiation fluxes perturbations [4,5,10,11].

In addition to the worldwide green-house gas effect, the local concentrations of short-lived climate forcers such as atmospheric aerosols play an important role in the fast Arctic warming [4,6]. For example, Ren et al. [4] reported that the combined total effect of sulfates and black carbon (BC) can explain approximately 20% of the Arctic warming since the early 1980s, causing a surface heating of 0.29 K. These aerosol species can act at different spatial and temporal scales by means of their direct and indirect effects [12,13], influencing the Arctic climate both as local forcers (local emissions or long-range transport from lower latitudes) and by changing poleward heat transport [5,11] through their forcing at midlatitudes. In particular, BC, the main light-absorbing carbon species, has been widely investigated in the last decade [14] because of its important contribution to climate change in the Arctic region [15]. For example, Sand et al. [9] found that BC locally emitted within the Arctic has an almost five times larger surface temperature response (per unit of emitted mass) compared to emissions at midlatitudes.

The sources of BC can be both anthropogenic (energy production, industrial activities, gas flaring, and domestic heating) and natural (forest fires, in case of the Arctic area, especially those in Canada and Siberia).

As well as BC, brown carbon (BrC; e.g., from forest fires) [16] and dust (e.g., high latitude dust) [17,18] also belong to the category of light-absorbing aerosols (LAA).

Once suspended in the atmosphere, LAA directly interact with solar radiation, absorbing it and exerting a positive atmospheric forcing (direct effect). This process releases energy (heat) to the surrounding air with a specific heating rate (HR) [12,19–26]. The LAA species are characterized by a significant spatial and temporal variation in their radiative effects, which can be larger than those of greenhouse gases on a regional scale [27,28]. Indeed, the HR can vary greatly in relation to location and local anthropogenic emission; e.g., Tripathi et al. [29] determined a BC induced HR of ~2 K/day in the urban area of Kanpur (northern India), while Ferrero et al. [30] found HR values between 0.5 and 1.5 K/day in Milan (Po Valley, Italy). Moreover, it has been found that dust aerosol can exert a positive atmospheric forcing with HR values up to ~2–3 K/day [31].

Among the LAA species, BC accounts for 70–90% of light absorption on a global scale, while BrC contributes for the remaining 10–30% [32,33] of the total LAA absorption.

The retreat of important glaciers (i.e., Himalayan) can also be influenced by LAA-induced HR [27,34], making its estimation even more important in the Arctic, where the aforementioned AA phenomenon is present. The surface temperature response depends on the altitude at which LAA heat the atmosphere [10,35,36], and it is known that LAA particles located near the ground (within the Arctic planetary boundary layer) cause a strong surface warming [35,37], in particular during spring and summer when solar radiation is highest. During spring, LAA (in particular BC) reach high concentration levels due to the Arctic haze phenomenon (transport from northern mid-latitudes). In summer, the polar dome retreats further north and transport of pollutants from lower latitudes drastically decreases; thus BC concentrations are lower in this season and local sources become more relevant: they emit BC directly within the Arctic dome, where it warms the surface due to the near-surface solar heating and the greater likelihood of surface deposition [9]. Indeed, LAA can also deposit on snow and ice, affecting their surface albedo and accelerating their melting [3,38].

Moreover, LAA can alter the atmospheric thermal structure, affecting the atmospheric stability and, consequently, the cloud vertical distribution and formation (semi-direct effect) [12,30,39].

It noteworthy that the LAA HR (once normalized for its atmospheric loading) can be larger in the Arctic with respect to other areas of the Earth because of the radiative contribution added to its absorption from highly reflective snow/ice surfaces [11], making its effect close to the surface even more important and its determination even more necessary. Most of the findings in the literature provide exclusively modelled or

experimental-based radiative-transfer computed values of aerosol-induced HR in the Arctic [40–43]. Treffeisen et al. [41] found local solar heating rate anomalies between 0.05 and 0.3 K/day (at 2 km altitude) during an Arctic haze event in March 2000, while Treffeisen et al. [42] calculated a daily mean HR value of 0.55 K/day at 0.5 km altitude (1.7 K/day if internal mixture, instead of external mixture, of the aerosols is considered) on 2 May 2006, due to an atypical transport of biomass burning aerosols from Eastern Europe. An older work by Porch et al. [43] reported an increase in the mean HR from 0.01 to 0.06 K/day in the lowest 1–5 km of the atmosphere under cloud-free Arctic haze conditions. These studies are thus focused on transport events, especially during spring. In a recent work, Donth et al. [40] showed a maximum daily mean BC HR, for spring simulations, of 0.1 K/day for the rather pristine Arctic Ocean, about an order of magnitude lower than the values at lower and tropical latitudes [29], while they found negligible HR values for the summer case. However, they focused only on the Arctic Ocean area, while other Arctic sites may be characterized by higher atmospheric LAA concentrations in summer due to local emissions.

Therefore, an experimental quantification of BC pollution (and more generally, of total LAA) and associated HR together with its distribution in the Arctic area are main scientific targets to unravel the future of the Arctic from a climate change perspective [3,44]. Thus, specific studies of these phenomena are required [45].

The Svalbard Archipelago is a key area in the Arctic, and it is affected by local sources such as ships and human settlements (e.g., Barentsburg, Longyearbyen, Sveagruva, Ny-Ålesund, and Hornsund). Their impact needs to be investigated in terms of the BC and atmospheric heating spatial variability at a local scale [44]. Furthermore, the Arctic BC emissions are expected to rise due to new oil and gas exploration and due to enhanced marine traffic in summertime, caused by the retreat of sea ice and increasing tourist and commercial cruises [3,9,10,46].

Nevertheless, apart from Ny-Ålesund, BC measurements in this area are sparse and not well spatially distributed, and the area's radiative forcing has never been calculated in a fully experimental manner. This study aims to add another contribution, focusing mainly on surface BC concentrations and their climate impact in terms of HR (only the direct effect was considered here) on the Arctic Ocean around the Svalbard Islands and within the anthropized fjords of Spitsbergen over two years (2018 and 2019) of summer campaigns. Considering the measuring technique applied in this study (Section 2), the mass of LAA is hereinafter expressed in terms of equivalent black carbon (eBC) [47] and, according to the literature, the corresponding HR is expressed in units of K/day. Moreover, eBC concentrations, radiation measurements, and the related forcing were accomplished entirely experimentally, avoiding the uncertainties and assumptions inherent in the climatic models: for instance, models that quantify BC climate impacts in the Arctic region usually tend to misrepresent the real BC concentration and the amplitude of its seasonal variation [44,48]; Sobhani et al. [49] and Sharma et al. [50] proved that models can overestimate the observed eBC concentrations during summer, while other studies [4] show an opposite behavior or have conflicting results as a result of the location.

In addition to eBC and its direct forcing, other aerosol properties have also been determined in order to better understand its role in climate modifications in this area. Indeed, the differences in chemical composition of Total Suspended Particles (TSP) and particle number size distribution between different areas/case studies were investigated too.

We describe the sampling campaigns and the aerosol measurements in Section 2, while the results and discussion follow in Section 3.

## 2. Materials and Methods

### 2.1. AREX Sampling Campaigns

Aerosol and radiation measurements were performed on board the S/Y Oceania (owned by the Institute of Oceanology Polish Academy of Sciences) during the AREX

(Arctic Expedition) research campaigns in the summer of 2018 and 2019. Both cruises involved transects which originated in Tromsø (north of Norway) in the second half of June (20 and 18 June for 2018 and 2019, respectively), then led to the Svalbard Islands and were completed on 20th August. The transects were similar in both cases (Figure 1): the ship spent most of the time in the Norwegian Sea (south of Svalbard) in the first part of the cruise, then it reached the northernmost point in the Arctic Ocean (passing the North of Svalbard). In the central part of the campaign, the ship entered the main fjords of Spitsbergen and finally headed south, sailing further west than the outward journey. The only difference between the two cruises was in Oceania sailing along the east coast of Spitsbergen in 2018 and going further west into the Greenland Sea, also reaching the Jan Mayen Island on the way back. These journeys enabled us to assess the impact of the following anthropic settlements: Tromsø, Longyearbyen, Barentsburg, Ny-Ålesund, Hornsund, Sveagruva, and Jan Mayen.

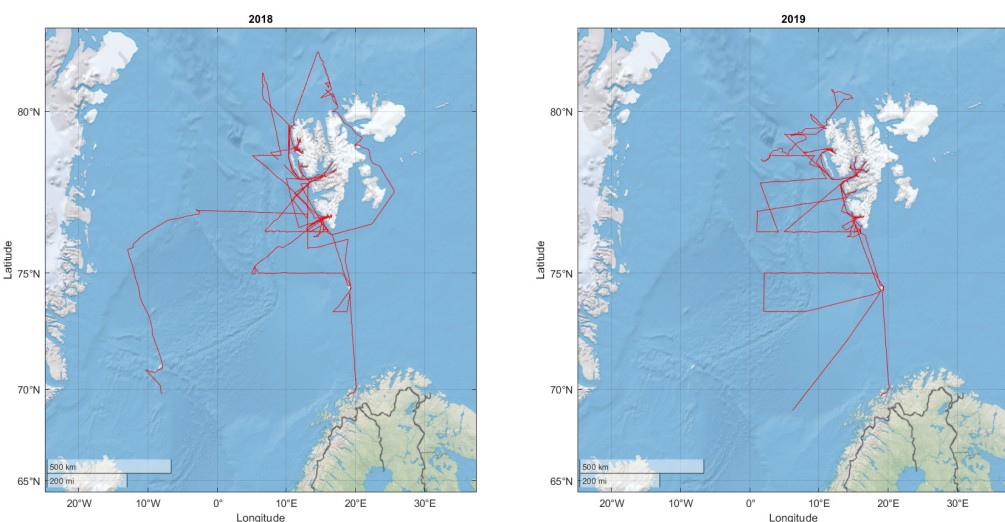

**Figure 1.** Route of the S/Y Oceania (red line) during the 2018 (**left**) and 2019 (**right**) AREX campaigns.

The shape of Oceania (Figure 2a) forces it to sail mainly with the wind from the bow, and the presence of three sailing masts (that allow reduced engine use) decrease the ship's impact on aerosol sampling. Instruments for sampling the total suspended particles (TSP) on filters and for measuring the PM concentration and solar radiation were installed on the measurement platform (balcony) on the main mast (Figure 2b). An Aethalometer AE33 (equipped with a dryer), to determine both the equivalent black carbon (eBC) concentrations [51] and the aerosol absorption coefficients, was installed in a sealed room (Figure 2b) to avoid any sea-spray damage. Aerosol sampling at 5 L min$^{-1}$ was ensured through a 2 m length sampling line out of the room equipped with a PM$_{2.5}$ cyclone.

## 2.2. Sample Collection, Extraction and Analysis

TSP samples were collected on quartz fiber filters (QFFs, 105 mm diameter, Whatman, Springfield Mill, Nottingham, UK) using a high-volume sampler (ECHO-PUF, TCR Tecora, Cogliate, Italy). This sampler was placed on the measurement platform, 10 m above the sea level (6 m above the deck of the vessel), closer to the bow. The sampling flow was set at 700 L/min for 2018 and 180 L/min for 2019. The lower flow during 2019 was due to a longer sampling time and the presence of a white filter underlying the sampled filter, in order to obtain a more homogeneous aerosol distribution on the filter surface for subsequent analysis. The average sampling time was 96 h over the ocean and 48 h in emission hotspots. QFFs were baked at 550 °C for 5 h before sampling to reduce impurities and stored in pre-cleaned (with Milli-Q water and acetone) aluminum foil at room temperature. After sampling, they were stored in the same aluminum foil in a refrigerated (−20 °C) and dark environment. Before each analysis, the filters were equili-

brated for 48 h at 35% RH and room temperature. Blank field filters were also collected to determine the detection limit (DL) of the different analytes. DL was calculated as the average value of all blank field filters plus three times the standard deviation [52]. The studied filters were all above the DL.

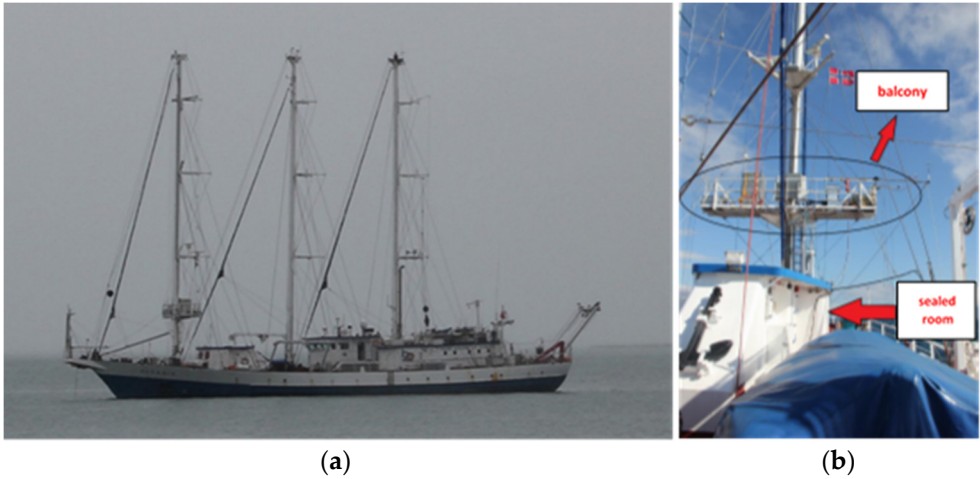

**Figure 2.** S/Y Oceania (**a**) and its balcony and tiny, sealed room (**b**) where the instruments for aerosol measurements were installed.

Filters were divided into two different punches, devoted to the following analyses: ion chromatography (IC, 15 mm punch) in order to infer water-soluble inorganic ions and total carbon analysis (TCA, 14 mm punch) in order to infer total carbon (TC) as well as organic carbon (OC). Uniformity of sampled aerosol on filters was demonstrated in a previous work [53].

### 2.2.1. Water-Soluble Inorganic Ions

Sampled filters were extracted once in 3.0 mL of ultra-pure water (18.2 MΩ cm resistivity, Milli-Q system; Millipore, Billerica, MA, USA) for 15 min in an ultrasonic bath (SONICA®; Soltec, Milan, Italy) to determine the water-soluble inorganic ions content. The extract was filtered (0.45 μm PTFE filters; Alltech, Nicholasville, KY, USA) in order to remove the insoluble fraction [13]. Five main cations ($Na^+$, $NH_4^+$, $K^+$, $Mg^{2+}$, and $Ca^{2+}$) and anions ($F^-$, $Cl^-$, $NO_3^-$, $SO_4^{2-}$, and $PO_4^{3-}$) were analyzed using an ICS-90 and ICS-2000 coupled ion chromatography system (Dionex, Sunnyvale, CA, USA), respectively (both fitted with a conductivity detector), equipped with an AS 3000 Autosampler (Dionex). Cation determination was accomplished through an Ion Pac CS12A-5μm analytical column (3 × 150 mm, Dionex) and an Ion Pac CG12A-5 μm (3 × 30 mm, Dionex) guard column, together with a CMMS III (4 mm, Dionex) electrolytic suppressor, using 0.4 M methansulfonic acid (MSA) as eluent at 0.5 mL min$^{-1}$. Anion determination was accomplished through an Ion Pac AS11 analytical column (4 × 250 mm, Dionex) and an Ion Pac AG11 guard column (4 × 50 mm, Dionex), together with a ASRS-300 (4 mm, Dionex) electrolytic suppressor [54], using 50 mM KOH as eluent at 0.8 mL min$^{-1}$. Quantification was made by using the external standard method. Cations and anions standard mixtures were prepared from mixing single liquid standards (1g L$^{-1}$; Fluka, Sigma-Aldrich, St. Louis, MO, USA) and then progressively diluting in Milli-Q ultra-pure water to obtain the calibration curves. All ions calibration curves have regression coefficients ($R^2$) > 0.995. The areas obtained from the average of the blanks were subtracted from the results of the different ions (above the DL) before converting them into concentration values.

### 2.2.2. TC and OC

Total carbon content on sampled filters was obtained via a thermal method, using a TCA-08 Total Carbon Analyzer (Aerosol Magee Scientific, Ljubljana, Slovenia) in offline mode: a filter punch was placed over the installed cleaned 47 mm filter and heated almost instantaneously in a small flow of 'analytical carrier gas' (ambient air). Thus, by means of complete combustion, all carbonaceous compounds are converted to $CO_2$, which is detected by a nondispersive infrared Licor $CO_2$ sensor. In this way, the $CO_2$ concentration over the carrier-gas baseline (the transient concentration of $CO_2$ exceeds the carrier-gas by far, due to the small internal volume of the system) is accurately measured and integrated to give the total carbon content of the sample. "Clean chamber test" and "Zero verification test" (to correctly determine the ambient-air baseline) were performed before each cycle of offline analysis. Then, organic carbon (OC) was obtained by combining the thermal method for TC determination with the optical method for eBC determination (TC-BC method [55]). Indeed, the eBC provided by the Aethalometer was averaged over the time periods of each TSP filter. These mean eBC values were subtracted from the total carbon ones, achieving the organic carbon content. Finally, OC was converted to organic matter (OM) using a factor of 2.1 for remote Arctic regions [10,56].

### 2.3. Black Carbon and the Related Absorption Coefficient Data

BC and absorption coefficient (at 7 wavelengths) measurements were determined using a dual-spot Aethalometer model AE33 (Aerosol Magee Scientific, Slovenia) with real-time loading compensation [57,58]. It was installed in a tiny room towards the bow of the ship (about 20 m from the stack, Figure 2b), inside a specially built recovery box to preserve it from aggressive and corrosive environmental conditions (cold, sea spray, high humidity, and ship movement). In order to obtain accurate data and protect the instrument from water condensation, a Sample Stream Dryer (Aerosol Magee Scientific) was used. It operates using a Nafion membrane to remove water vapor from the sample stream via permeation to an outer space maintained at a low absolute pressure by means of an external vacuum pump. A $PM_{2.5}$ cyclone at the end of the sampling line allowed us to cut out aerosols greater than 2.5 μm, while an insect and water trap allowed us to further prevent the entry of rain or condensed water. The sampling line extended 2 m from the roof of the small room via a metal hollow pole, and the cyclone was positioned 6 m above the sea surface, thus avoiding most of sea waves. The time-base period was set with a 1 min resolution.

The Aethalometer is based on an optical method using the estimations of attenuation (ATN) of transmitted light. The absorption coefficient, $b_{abs}$, is calculated from the attenuation coefficient, $b_{ATN}$, as follows:

$$b_{abs} = \frac{b_{ATN}}{C * (1 - k * ATN)} \tag{1}$$

where *k* and *C* are the loading effect parameter and the multiple scattering optical enhancement, respectively. The loading effect parameter, which compensates for the non-linear loss of the measurement sensitivity as the load on the filter spot increases, changes with time of the year and position [59], but in the dual spot Aethalometer, *k* is inferred at the same time resolution as the measurement itself by means of a real-time compensation algorithm [57]. *C* describes the enhancement of the optical path (and consequently of the absorption) of the aerosol on the filter due to the multiple scattering of light within the filter fibers when the filter is relatively unloaded [60,61]. The loading effect parameter used by default for T60 tape (TFE-coated glass filter, Pallflex "Fiberfilm" T60A20) in the AE33 is 1.57. However, the measurements were rescaled by a constant *C* factor of 4.1, recently found in the Arctic area at the Zeppelin station (Ny Ålesund, Svalbard) [62]. BC ambient concentration was obtained using the reference measurement at 880 nm on spot 1 (the one

with the higher flow) and the apparent mass attenuation cross-section (*MAC*) value at the same wavelength:

$$BC = \frac{b_{abs}}{MAC} \tag{2}$$

*MAC* at 880 nm is equal to 7.77 m$^2$ g$^{-1}$ [57,63] and is provided by the manufacturer. Below, according to the literature [10,22,51], as stated in the introduction, we will refer to the BC concentration calculated with Equation (2) as "mass-equivalent-BC" (eBC), as it depends on the whole amount of LAA in the atmosphere and because *MAC* depends on the dimension, morphology, composition, and mixing of aerosol particles, as well as on the filter material, relative humidity, and airflow [14,64,65]. eBC data (calculated with standard pressure and temperature values of the Aethalometer) was also corrected using actual measured weather data of temperature, pressure, and relative humidity. Moreover, eBC data were treated with a 30 min moving mean, which corresponds to a maximum latitudinal spatial resolution of 9 km, considering the average speed of the ship without stopping. It was used for smoothing out the instrumental noise of the Aethalometer in the Arctic area, where the concentrations are particularly low. High concentrations due to the possible interference of the vessel exhaust were eliminated by applying a vessel speed/apparent wind (180° ± 40°) filter (it can occur during stops and starts and when wind is coming from the stern) before the treatment with the moving average. Figure 3 shows a day with apparent southerly wind and the effect of the wind filter on eBC concentrations. However, due to the Oceania's tendency to sail headwind and the greater height of the chimney above the sampling line, there was only a small percentage of data (less than 2%) removed from the total.

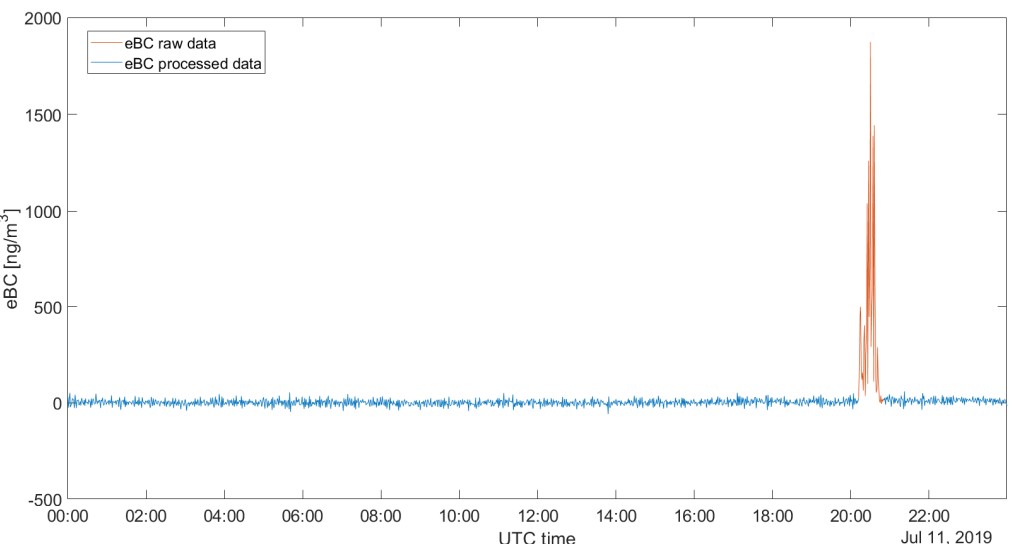

**Figure 3.** eBC concentrations on 11 July; red line represents the original data, while the blue line represents eBC data after application of the wind filter.

*2.4. Solar Radiation Measurements*

Solar radiation measurements are necessary to experimentally determine the aerosol heating rate (HR) as shown in the next paragraph. Therefore, one pyranometer and one spectroradiometer were installed on the balcony of the S/Y Oceania. SPN1 pyranometer (Delta-T Devices, Cambridge, UK) measures short wave radiation (W m$^{-2}$) between 400 nm and 2700 nm and returns 3 outputs: global solar radiation, diffuse radiation and sunshine status. It was coupled with a The GP1 Data Logger (Delta-T Devices). The hexagonal arrangement of the 7 radiation sensors and the hemispherical shadowmask, with specific areas cut away, allows the logger to always have at least one sensor exposed to direct solar radiation and at least one sensor completely shaded, while all sensors receive about half of the diffuse radiation equally (assuming that the diffuse sky radiance is isotropic) [66–68]. Thus, the SPN1 can measure the diffuse radiation without moving parts,

such as shadowbands or solar trackers. The direct solar radiation can be easily obtained by subtracting the diffuse component from the total. The main problem on moving platforms, such as aircrafts and ships, is the tilt of the instrument and the consequent error induced in the measurement of the direct component. The diffuse component, which is considered isotropic, is not affected by this problem. Therefore, the direct component must be corrected in order to obtain a correct global radiation, given by the sum of the new direct radiation and the diffuse one (unvaried). To correct the SPN1 direct radiation, the method shown by Wood et al. [67] and derived from Long et al. [66] was used. To apply this method, it is necessary to know the position of the sun (solar zenith and azimuth angles) and the angle of incidence on the tilted instrument plane, which is calculated from the ship's orientation values (pitch, roll, and yaw from an Ellipse-E Inertial Navigation Sensor, by SBG systems).

A RoX (reflectance box) spectroradiometer (JB Hyperspectral Devices, Düsseldorf, Germany) was also installed on the measurement platform of S/Y Oceania. It is based on the Multiplexer Radiometer Irradiometer (MRI) described in the work of Cogliati et al. [69], and it collects incident and reflected radiation in the visible/near infrared region (300–950 nm) by exploiting the Flame spectrometer from Ocean Optics (Orlando, FL, USA). The raw data of the RoX was processed using the open-source R-package provided by JB company (Düsseldorf, Germany). Moreover, the RoX used during the campaigns was customized by the manufacturer according to specific requirements: a gimbal and a rotating shadowband had been added. The gimbal was used to compensate for ship motion, so that it was not necessary to correct data afterwards, as was done for SPN1. The shadowband rotates 5 degrees per minute to cover, step by step, all 180 degrees of the upwelling sensor's field of view. Thus, the upwelling sensor is obscured at regular time intervals. When the sensor is uncovered, global radiance is measured; diffuse radiance is measured instead when the band shades it. This improvement allows us to separate the diffuse radiation from the global one and also obtain the direct radiation by subtracting the direct component from the total (as previously done in Ferrero et al. [70]).

Both SPN1 and RoX acquired absolute and spectral data each minute, the same time resolution as the Aethalometer.

*2.5. Heating Rate Determination*

By means of LAA absorption coefficients and the radiation measurements described above, it was possible to determine the surface atmospheric heating rate related to LAA, using the methodology developed by Ferrero et al. [70]. The novelty of this method is that it allows us to assess, in a completely experimental manner, both the radiative power density absorbed into a near-surface atmospheric layer and the consequent HR. Instead, most of the studies concerning radiative forcing are conducted through a hybrid [23,29,40,71,72] (experimental measurements of LAA properties coupled with radiative transfer calculations) or purely modeling [6,11,39,73] approach, affected by uncertainties related to input parameters [74], such as modelled aerosol concentrations and optical properties, and to the frequent assumption of clear-sky conditions. Thus, this experimental method enables us to determine the HR in any sky condition. Another advantage is the opportunity to apportion the HR in relation to of LAA species and sources and different radiation components. This HR computation methodology starts from the consideration that aerosol measurements are usually expressed as mass (or number of particles) per unit volume of air. Therefore, it is more useful to define HR in terms of thickness of the atmospheric layer ($\Delta z$) rather than pressure variation ($\Delta P$), as follows:

$$HR = \frac{\partial T}{\partial t} = \frac{1}{\rho C_p} \frac{\Delta DRE}{\Delta z} \qquad (3)$$

The last term ($\Delta DRE / \Delta z$) is the so-called *ADRE* (absorptive direct radiative effect) [23], which is the radiative power absorbed by the aerosol for unit volume of the atmosphere

and whose unit of measure is (W m$^{-3}$), consistently with aerosol concentrations. In fact, $\Delta DRE$ is the instantaneous radiative power density absorbed by the aerosol (W m$^{-2}$) [26] and is given by the difference between the instantaneous aerosol direct radiative effect ($DRE$; W m$^{-2}$) at the top and the bottom of an atmospheric layer of thickness $\Delta z$. Instead, $\rho$ is the air density (kg m$^{-3}$), while $C_p$ (1005 J kg$^{-1}$ K$^{-1}$) is the isobaric specific heat of dry air. From these considerations and using the following definition for the amount of radiation absorbed by the aerosol within an atmospheric layer (near surface layer for this study) of thickness $\Delta z$ on which an $n$th kind (direct, diffuse, or reflected) of monochromatic radiation ray $F_{n(\lambda,\theta)}$ of wavelength $\lambda$ arrives with a zenith angle $\theta$ [26,70]:

$$\Delta DRE_{n(\lambda,\theta)} = F_{n(\lambda,\theta)} \left(1 - \omega_\lambda\right) \left(1 - e^{-\tau_\lambda/\mu}\right) \tag{4}$$

where $\omega_\lambda$ is the single scattering albedo of the aerosol present in the atmospheric layer, $\tau_\lambda$ is the aerosol optical depth, and $\mu$ is the cosine of $\theta$; considering also the definitions of $\omega_\lambda$ and $\tau_\lambda$ (refer to Ferrero et al. [70] for a more detailed methodology), the $ADRE$ for this atmospheric layer can be written as follows:

$$ADRE_{n(\lambda,\theta)} = \frac{dDRE_{n(\lambda,\theta)}}{dz} = \frac{F_{n(\lambda,\theta)}}{\mu} \, b_{abs(\lambda)}. \tag{5}$$

Therefore, considering Equation (5), it is clear that the $ADRE$ and the consequent HR can be calculated using the LAA absorption coefficients (given by the Aethalometer) and radiation measurements. The HR is thus calculated by integrating the previous equation for all wavelengths and incident angles and taking into account all the three components of shortwave radiation:

$$HR = \frac{1}{\rho C_p} \sum_{n=1}^{3} \int_\theta \int_\lambda \frac{F_{n(\lambda,\theta)}}{\mu} \, b_{abs(\lambda)} \, d\lambda \, d\theta \tag{6}$$

In this work, the recent experimental HR method, which is independent of the thickness of the atmospheric layer considered (because $ADRE$, i.e., the vertical derivative of DRE, is directly used in the computation of HR instead of DRE), was applied to the surface, but the results obtained can be representative of the mixing layer height when there are no specific gradients of LAA concentrations in the lower atmosphere [23,70,71]. As already assumed in relation to the radiation measurements (SPN1 and RoX), $ADRE$ is calculated considering the diffuse radiation isotropic. HR was also computed over the broadband range of SPN1 (using the Angstrom exponent of $b_{abs(\lambda)}$), but the HR reported here is that obtained only in the range covered by the wavelengths of the Aethalometer (370–950 nm).

### 2.6. CPC and LAS Measurements

In order to obtain particle number concentrations, a TSI Condensation Particle Counter (CPC, Model 3771) and a laser aerosol spectrometer 3340 LAS (TSI Inc., Shoreview, MN, USA) were also installed on the balcony inside a special box to protect the instruments. The CPC measures the total aerosol concentration with a nominal minimum detectable particle size ($d_{50}$) of 10 nm at an aerosol flow rate of 1.0 L per minute, over a concentration range from 0 to $10^4$ particles per cubic centimeter. This device was evaluated and tested inter alia by Takegawa and Sakurai [75].

The LAS counts aerosol particles from a diameter size of 0.09 μm up to 7.5 μm in 99 channels. Thanks to the application of wide-angle optics and intracavity He-Ne laser, the LAS ensures a wider measurement range. Device was tested in aerosol measurements in many field observations [76–79]. A PM$_{10}$ cyclone was mounted at the LAS inlet to prevent the entry of water.

To keep the same temporal resolution as the other real-time measurements, we set up the measurement time step at 1 min. Finally, we applied the same vessel speed/apparent wind filter that we used to clean the eBC data from the possible influence of the ship.

*2.7. Data Analysis Strategy*

For the purpose of this work, which aims at the determination of the climatic impact (in terms of heating rate) of the anthropic settlements in the Arctic, we divided the collected real-time database into the following case studies:

1.  *Arctic Background*, which represents the aerosol concentrations on the sea in pristine conditions, i.e., the concentrations found at steady state after dilution and deposition phenomena and in the absence of evident local emissions and transport phenomena.
2.  *Anthropic fjords*, i.e., Spitsbergen fjords characterized by the presence of human settlements; these are Hornsund fjord (where the Polish Polar Station is located), Kongsfjorden (where Ny-Ålesund settlement is located, also including Krossfjord due to its continuity with Kongsfjorden in this analysis), Isfjorden (where Pyramiden, Barentsburg, and especially Longyearbyen are located, including also the secondary fjords branching off from it), and Van Mijenfjorden (including Sveagruva).
3.  Other human *hot spots* in the Arctic area, i.e., Tromsø (the main city of northern Norway) and the Jan Mayen Island, also characterized by human presence.
4.  *Local Settlements Pollution Effect* (LSPE), i.e., the diffusion of the anthropogenic impact detectable in the sea area around the anthropic fjords and not within the fjords themselves (identified thanks to the proximity to the fjords and the use of back-trajectories and wind direction).
5.  *Long-Range Transport Events* (LRTE), characterized by aerosol concentrations, particularly BC, in the open sea clearly above the background, whose origin (Northern Europe and Russia) has been traced through air mass back-trajectories. These were computed using the on-line version of the hybrid single particle Lagrangian integrated trajectory model (HYSPLIT) developed by the National Oceanic and Atmospheric Administration Air Resource Laboratory (NOAA, https://www.arl.noaa.gov/HYSPLIT/, accessed on 22 November 2023). Figure 4 shows a seven-day back trajectory, obtained with a 6 h resolution and determined at 100 m, 500 m, and 1000 m above the sea level, for a Long-Range Transport Event. Some other back-trajectories are reported in the Supplementary Material (from Figures S1–S4).

The analysis of the different case studies includes both years (2018 and 2019). In only the case of Jan Mayen and Van Mijenfjorden, the analysis involved just one year, 2018 for Jan Mayen and 2019 for Van Mijenfjorden, due to the fact that they were reached by S/Y Oceania only in these years.

Instead, regarding the off-line analyses, the comparison includes the background and the two main hotspots, i.e., Tromsø and Longyearbyen. Other areas/case studies are not involved in the chemical discussion. Indeed, the ship spent less time continuously in the various areas/case studies than the time needed for sampling, except for background areas, Tromsø, and Longyearbyen. As a result, there are no filters representing solely the other specific sites, without being contaminated by the background or other areas (they reflect instead mixed conditions). For this reason, we only considered 8 filters out of the 16 sampled during the two campaigns in the Arctic area. However, the main aim of this work is to link the high-temporal resolution measurements to the HR and thus we only used the chemical composition to introduce macroscopic differences, while real-time data allowed a much more detailed analysis.

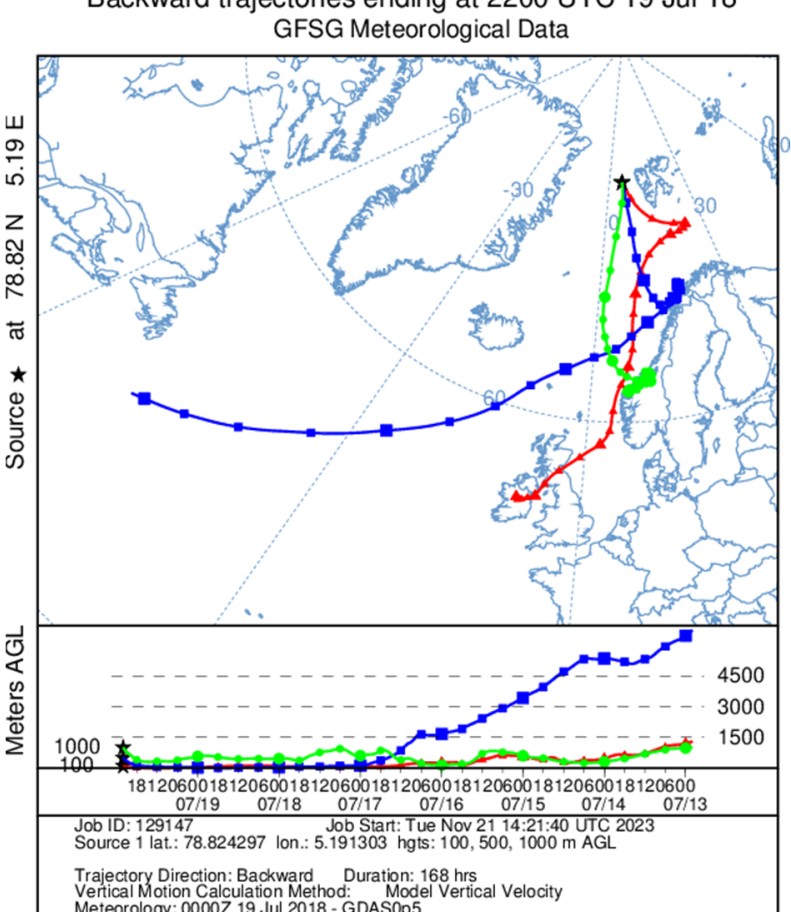

**Figure 4.** Seven−day air mass back−trajectories derived using HYSPLIT for the LRTE that occurred on 19 and 20 July 2018. Back-trajectories ending at 22:00 UTC 19 July 2018 at 100 (red), 500 (blue), and 1000 (green) m above ground level.

### 3. Results and Discussion

The study outcomes presented below involve results, separating the off-line outputs (i.e., aerosol chemical composition) from the real-time ones.

High-time-resolution measurements were grouped in eBC and particle concentrations (CPC and LAS) on one hand and HR results on the other hand, and they were divided according to the areas/case-studies already presented in the methodology section: Arctic background, i.e., pristine sea conditions between 69° and 81° N; Svalbard fjords, characterized by the presence of human settlements (here called anthropic fjords); Tromsø and Jan Mayen Island, i.e., the other two arctic hotspots outside Svalbard reached by the ship; Local Settlements Pollution Effects; and Long Range Transport Events. Among the anthropic fjords, Van Mijenfjorden is a special case, as we will see also from the results in the next paragraphs. In fact, it is the only fjord that was covered for half of its length without reaching the actual settlement, Sveagruva (see Figure S5 in Supplementary Material). Most of the sampling time was classified as background (38.0%), while 32.2% and 4.8% of the time was spent in the anthropic fjords (14.8% Isfjorden, 8.8% Hornsund, 8.2% Ny-Ålesund, and 0.4% Sveagruva) and in the other hotspots (3.4% Tromsø and 1.4% Jan Mayen), respectively. Then, 2.3% and 1.5% of the time was classified as LRTE and LSPE, respectively, while the remaining measurements (21.2%) were not used because of uncertainties in the classification or because the data were deleted due to the influence of the vessel itself. The number of real-time valid data at 1 min time resolution for each areas/case studies is shown in Table S1. Real-time, 1 min time resolution, valid data range from 549 in Van Mijenfjorden

to 47654 for the Arctic background. The value of Van Mijenfjorden is particularly low because it is a special case (as explained above): it is the fjord where the ship spent the least time and it was covered only for half of its length. For all other areas/case studies, there are more than 2000 (1 min time resolution) valid data.

### 3.1. Chemical Composition

Water-soluble inorganic cations ($Na^+$, $Ca^{2+}$, $Mg^{2+}$, $K^+$, and $NH_4^+$) and anions ($Cl^-$, $SO_4^{2-}$, $NO_3^-$, $PO_4^{3-}$, and $F^-$) and organic matter were detected on the filter samples. From the results, shown in Figure 5, the difference in chemical composition between the Arctic Ocean background and the two most important emissive hotspots (Tromsø and Longyearbyen) is clearly evident. The percentages shown in the pie charts are expressed in terms of relative mass. As regards the Arctic Ocean, since it is a typical marine aerosol, the most abundant ions were $Cl^-$ and $Na^+$ with mean concentrations of $1246.7 \pm 455.8$ ng/m$^3$ and $761.9 \pm 245.3$ ng/m$^3$, respectively. The average relative contribution of eBC, which has its only primary origin from incomplete combustion, was minimal (about 0.2%; mean concentration of $5.5 \pm 0.2$ ng/m$^3 \approx$ half of the value found by Massling et al. [80] at the Villum Research Station, North Greenland, but within the range of its confidence interval), indicating a pristine environment in the absence of evident transport phenomena, with a low presence of anthropogenic or natural combustion products, especially fossil fuel.

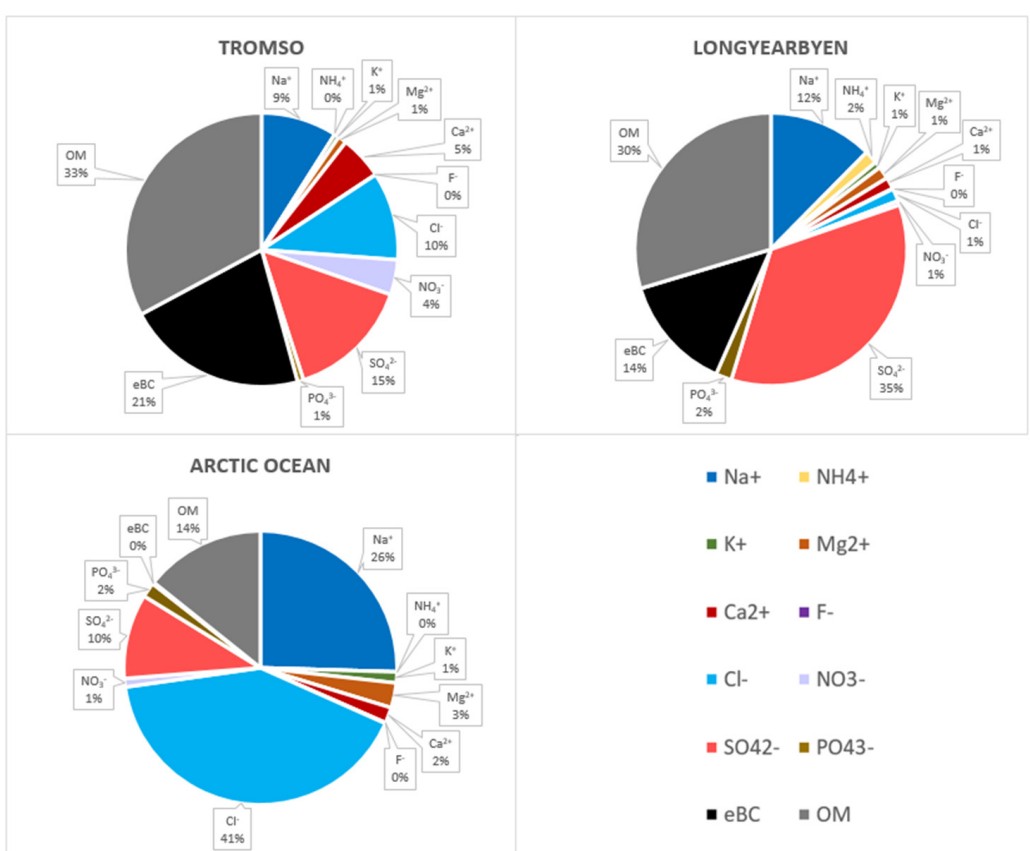

**Figure 5.** Aerosol chemical composition (relative percentages) for two important emission hotspots in the Arctic area (Tromsø and Longyearbyen) and for the Arctic Ocean around the Svalbard Islands.

Organic matter was the third most abundant compound among those analyzed, suggesting the possible presence of important biological emissions from the sea. The organic fraction, indeed, accounted alone, on average, for 95.5% of the total carbon content. The OM mean value was $383.3 \pm 79.9$ ng/m$^3$. Sulfates concentrations were also important ($293.3 \pm 84.4$ ng/m$^3$) because the presence of $SO_4^{2-}$ is due to the contribution of several factors: primary emissions of sea spray, secondary formation from

DMS precursor, emitted by phytoplankton activities (both represent a natural marine origin), and also long-range transport from the continents favored by its stability in the atmosphere (anthropogenic origin from precursors emitted by fossil fuel combustion). The other chemical species analyzed were less significant in terms of concentration. $NO_3^-$ was remarkably less abundant ($26.9 \pm 7.0$ ng/m$^3$) than $SO_4^{2-}$ because it has only a secondary (photochemical) origin from anthropogenic precursors (e.g., NOx) present on land (except for ship emissions) and it is less stable in the atmosphere, tending thus to volatilize during long-range transport. Its values are slightly lower (but in the same order of magnitude) than those reported by Xu et al. [81] for the Southern Ocean. The mean concentration of $PO_4^{3-}$ was $46.7 \pm 21.2$ ng/m$^3$. The fluoride ion was negligible (relative mean contribution to the total concentration less than 0.1%), while $NH_4^+$ was below the detection limit in any of the considered samples from the Arctic Ocean (this is consistent with the fact that its main source is related to long-range transport from continents, which is more relevant in the Arctic haze season, and with the negligible concentrations also found by Ferrero et al. [10] during spring). The remaining ions ($Ca^{2+}$, $Mg^{2+}$, and $K^+$) are part of the major ions in seawater, together with $Cl^-$ and $SO_4^{2-}$. Sea salt contribution to their observed concentrations can be determined according to Mihalopoulos et al. [82]:

$$SS_X \left[ \mu g/m^3 \right] = \frac{X}{Na^+} * Na_{measured}^+ \left[ \mu g/m^3 \right] \tag{7}$$

where X is the selected species and $X/Na^+$ is the mass ratio of its concentration to that of $Na^+$ in sea water [83]. The $Mg^{2+}$ concentration was $88.2 \pm 29.5$ ng/m$^3$ and it derived almost completely from sea salt aerosol ($99.8 \pm 0.1\%$; Figure 6). As expected, the chlorine also originated completely from sea spray. Calcium, instead, was the ion with the lowest marine origin ($53 \pm 1.7\%$) due to its predominantly crustal origin (e.g., material transported from Greenland and Iceland can be enriched in gypsum [17,84]); its mean concentration was $53.9 \pm 13$ ng/m$^3$. Almost three quarters ($72.6 \pm 1.3\%$) of the $K^+$ concentration ($38.8 \pm 12.1$ ng/m$^3$) derived from sea salt, while the non-sea salt (nss) part can have both crustal and forest fire origins. Finally, $65.2 \pm 2.3\%$ of $SO_4^{2-}$ was due to primary emissions from the sea. Nss-$SO_4^{2-}$ can derive from secondary emissions, both marine (DMS) and anthropogenic (coal and petroleum derivatives), as explained above. The Nss-$SO_4^{2-}$ mean value was $102.1 \pm 29.4$ ng/m$^3$, very similar to that reported at Villum Research Station (North Greenland) by Massling et al. [80]. Considering that $SO_4^{2-}$ was the most plentiful ion after $Cl^-$ and $Na^+$ and also considering its high nss-percentage over the sea during summer (i.e., not during the Arctic haze period, but when the biogenic fraction is more relevant [85], especially taking into account the positive summer trend over the past decades due to temperature increase and the related loss of sea-ice [85,86]), it is possible that a significant part of the sulfate comes from marine biogenic emissions, according to the high value of OM found and to Udisti et al. [87] too.

By comparing the two hotspots with the Arctic Ocean, the relative percentages of $Na^+$ and $Cl^-$, typical ions of sea salt aerosols, decreased considerably, and there was a parallel increase in the anthropogenic footprint, especially eBC, whose concentrations became $567.9 \pm 25.8$ ng/m$^3$ and $116.7 \pm 15.7$ ng/m$^3$ for Tromsø and Longyearbyen, respectively. The OM value in Longyearbyen was slightly lower than the Arctic background ($251.3$ ng/m$^3$), even if its origin was probably different and more linked to combustion rather than to biogenic activity, while in Tromsø it was definitely greater ($868.9$ ng/m$^3$) and largely due to anthropogenic activity. Total sulfate concentrations were similar for Tromsø ($389.7$ ng/m$^3$) and Longyearbyen ($296.7$ ng/m$^3$), but nss-$SO_4^{2-}$ sharply increased in both cases ($329.7$ and $270.3$ ng/m$^3$, respectively), with the minimum sea salt percentage (8.9%) reached in the main Svalbard human settlement, showing a fall in primary emissions from the sea. Nss-$SO_4^{2-}$ concentrations in the two hotspots were likely due more to local anthropogenic emissions than to biogenic marine activity (in contrast to Arctic Ocean samples), as in Longyearbyen. Indeed, it is not a remote station

but a hotspot located on land, where fuel (in particular coal and diesel) is burned for the production of energy and for heating. This agrees with the decrease in sea salt sulfate percentage, with the increase in nss-$SO_4^{2-}$ absolute value, with the sharp rise in eBC concentrations, and with the literature too: measurements in Ny-Ålesund showed that anthro-$SO_4^{2-}$ remains the principal component of nss-$SO_4^{2-}$ during summer, although it decreases compared to spring when the Arctic haze phenomenon is present [85,87]. The $NO_3^-$ value was significant in Tromsø (109.8 ng/m$^3$), as expected due to its location and anthropogenic sources, especially vehicular traffic. As in the Arctic Ocean, $Cl^-$ and $Mg^{2+}$ ions also originated exclusively from sea salt in the two human settlements. As expected, due to its source, $Mg^{2+}$ concentrations decreased (26.9 and 12.7 ng/m$^3$). The Ss-$K^+$ percentage decreased slightly, together with its concentrations, which became negligible. There was also a further reduction in the $Ca^{2+}$ sea salt percentage, especially in Tromsø (only 6.7%); the $Ca^{2+}$ absolute value was lower in Longyearbyen (11.8 ng/m$^3$) than in the Arctic Ocean, while it increased significantly in Tromsø. $F^-$ remained negligible, and the $PO_4^{3-}$ concentration was lower in both cases (18.9 and 16 ng/m$^3$). $NH_4^+$ was detected only in Longyearbyen (13.4 ng/m$^3$) and accounted for 2% of the total relative mass.

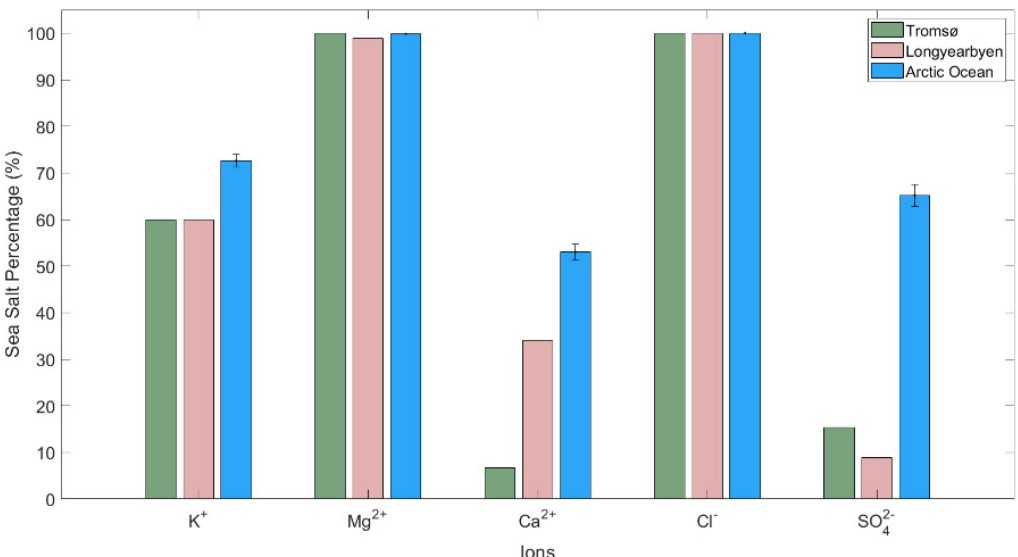

**Figure 6.** Sea salt origin for the analyzed ions also present in the sea water as major constituents.

The differences between the two hotspots are firstly due to their different size; one is the main city of the north of Norway, while the other is a smaller settlement, although its emissions are important in the Arctic context. Therefore, absolute concentrations are generally higher in Tromsø. Looking at the relative percentages (Figure 5), differences in the anthropogenic derivatives (eBC and $SO_4^{2-}$) can be probably ascribed to the use of different fuels. Indeed, coal is still a primary fuel for the local production of heat and electricity in Svalbard, where there are two coal-fired power plants in Longyearbyen and Barentsburg [88–90]. This agrees with the fact that nss-$SO_4^{2-}$ concentrations were similar in the two hotspots, although almost all the other investigated species were decidedly greater in Tromsø and although it was not the Arctic haze season (thus highlighting the importance of local sources).

It is clear that the aerosol chemical composition (Figure 5) can be highly impacted by local sources, requiring a deeper investigation using high-time-resolution measurements as detailed in the next section.

### 3.2. Real-Time Measurements

#### 3.2.1. eBC and Particle Concentrations

As stated in the introduction section, the impact of local eBC sources in the Arctic summer is crucial: they emit BC directly within the Arctic dome, where it causes a strong surface warming [9,35]. Therefore, we present the eBC and particle concentrations on the Arctic Ocean around the Svalbard Islands and within their anthropized fjords. Figure 7 shows the boxplots of eBC measurements during the 2018 and 2019 AREX campaigns, divided by the areas/case-studies previously explained: the central mark indicates the median and the bottom and top edges of the box indicate the 25th and 75th percentiles, respectively; the whiskers extend to the most extreme data points not considered outliers, while the outliers are not plotted. The mean (dashed line) and the related 95% confidence interval are also represented. The numbers above the top edge of the box point out the maximum value (i.e., the maximum outlier) reached. Figure S6 shows the same boxplots, but on a logarithmic scale, also including Tromsø in the main plot (this figure is useful for comparing Tromsø with the other arctic sites, but the differences between them are flattened by the logarithmic scale).

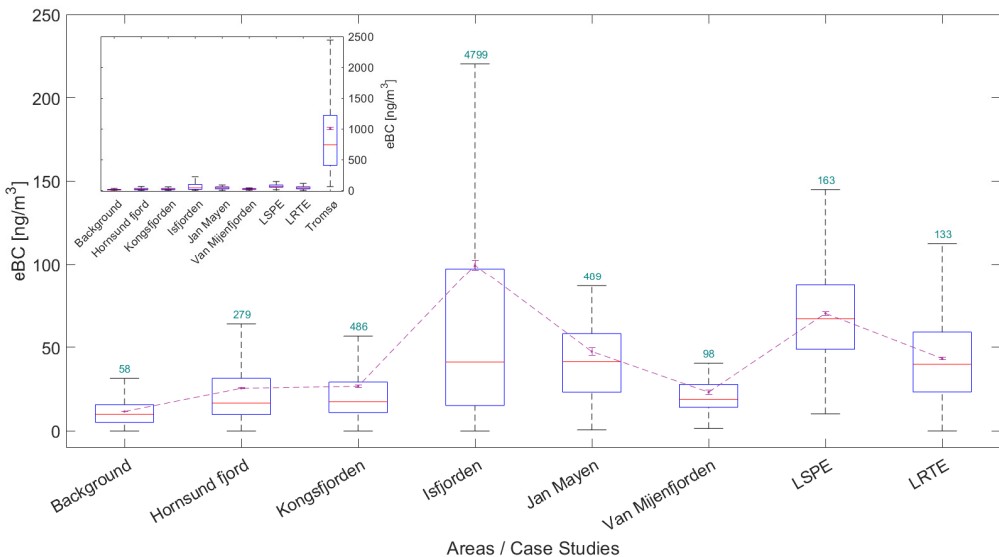

**Figure 7.** eBC boxplots for the areas/case-studies considered. The central mark indicates the median, while the bottom and top edges of the box indicate the 25th and 75th percentiles, respectively; the whiskers extend to the most extreme data points not considered outliers (outliers are not plotted). The mean (dashed line) and the related 95% confidence interval are also represented. The cyan numbers above the top edge of the box point out the maximum value (i.e., the maximum outlier) reached.

All the measurements were carried out in summer, when, on one hand, eBC concentrations are lower due to reduced transport from mid-latitudes and increased wet-deposition efficiency [91,92], and on the other hand, solar radiation is at its maximum, triggering the climatic effect of the light-absorbing aerosols. Ferrero et al. [10] reported homogenous vertical profiles of eBC in summer if fresh local emissions were not present. Under these conditions (i.e., background conditions), surface concentrations can be considered representative of at least the first km of the atmosphere.

Background (BG) conditions, which correspond to pristine conditions over the sea, showed an average value of $11.7 \pm 0.1 \, \text{ng/m}^3$, similar to the median value of $9.9 \, \text{ng/m}^3$, and a maximum value of $58.4 \, \text{ng/m}^3$. These results agree with the summer–autumn concentrations, generally <20 $\text{ng/m}^3$, as reported by Eleftheriadis et al. [15] for the period 1998–2007, and with the EC summer concentration of $11 \, \text{ng/m}^3$ (from 1990 to 1992) reported by Heintzenberg et al. [93] at the Zeppelin station. They are also similar to the annual-mean surface value of observed eBC ($13 \, \text{ng/m}^3$) for the period 2009–2015, described by Matsui et al. [94]. Zeppelin station is located near Ny-Ålesund at an

elevation of 464 m a.s.l. and can be considered a remote site, due to very little influence from the human settlement, and thus is compared here to background conditions. Indeed, despite measurements at the Zeppelin site being generally considered to represent free-troposphere environment, there is evidence that it is frequently representative of Boundary Layer conditions during late spring/summer [15,95]. Our BG results are similar also to summer eBC concentrations ($18 \pm 72$ ng/m$^3$) recently found at the Bely Island, in western Siberia [3] (although they are characterized by a much wider confidence interval). Obviously, BG concentrations were lower than all the other cases examined, also in accordance with the chemical composition of TSP (see previous results section), indicating a low presence of anthropogenic pollutants and an origin mainly linked to natural marine emissions.

eBC concentrations in Tromsø were off the charts compared to the other cases (Figure 7, inset), as expected considering its size and harbor activity, which makes it the main city of northern Norway and an important emission hotspot at the gateway to the European Arctic region. The mean eBC concentration in Tromsø was $1004.9 \pm 25.1$ ng/m$^3$, with peaks up to 7568.4 ng/m$^3$ measured in the harbor of the city. These results, once again, agree with the results reported in the previous section, which showed the highest relative percentages of eBC (and also $NO_3^-$) in Tromsø, as well as high percentages of OM and $SO_4^{2-}$. Jan Mayen is the other Arctic hotspot outside Svalbard reached by S/Y Oceania, and its mean concentration ($47.4 \pm 2.2$ ng/m$^3$) was in the same order of magnitude as the other polluted cases considered in Svalbard (with maximum values very similar to those found in Ny-Ålesund).

Among the Svalbard anthropized fjords, Kongsfjorden, Hornsund fjord, and Van Mijenfjorden showed analogous average concentrations: $26.7 \pm 0.6$ ng/m$^3$, $25.5 \pm 0.5$ ng/m$^3$, and $23.1 \pm 1.4$ ng/m$^3$, respectively. These three fjords are characterized by the presence of only one small human settlement. The mean value of Kongsfjorden included lower values measured in the deepest branches of the fjord (and also in the adjacent Krossfjord), and higher values (up to 486 ng/m$^3$) measured near the settlement and the harbor of Ny-Ålesund. Maximum values in Kongsfjorden agree with the mean surface value found by Ferrero et al. [10] in Ny-Ålesund for the vertical profiles affected by ship emissions ($319 \pm 14$ ng/m$^3$). Our average value is similar to that reported by Gogoi et al. [96] for the summer season ($19.5 \pm 6.5$ ng/m$^3$) at Gruvebadet observatory and to the average eBC observed during the period 2005–2018 (16–20 ng/m$^3$) [44]. Gruvebadet is located about 1 km southwest of the village of Ny-Ålesund. Its location guarantees generally low influence of local pollution sources given the prevailing southerly winds [97], but there is also evidence of frequent episodes of high absorption coefficient values, likely due to the local emission sources constrained in the lower layers of the troposphere [44]. Hornsund fjord is characterized by the presence of the Polish Polar Station, and its mean and median eBC concentrations were almost identical to Kongsfjorden, while the maximum values were lower, probably due to lesser naval traffic compared to Kongsfjorden. As previously mentioned, Van Mijenfjorden is a separate case because it was not entirely sailed through (unlike the other anthropized fjords) and the settlement of Sveagruva (located at the end of the fjord) was not reached. Usually, concentrations in the anthropized fjords are the result of measurements taken both in the harbor and near the human settlement of the fjord and in remote areas of the fjord. This could be the reason why maximum values in Van Mijenfjorden were lower than in Kongsfjorden and Hornsund. Moreover, less time was spent inside it than in other fjords.

Isfjorden is characterized by the presence of Pyramiden, Barentsburg, and especially Longyearbyen, the main human settlement and harbor on Svalbard. Consequently, its concentrations were particularly high for the Arctic area (only lower than those measured in Tromsø): it presented the highest mean eBC value ($99.4 \pm 3.1$ ng/m$^3$) among all the cases analyzed, except for Tromsø. The median value (41.2 ng/m$^3$) was similar to Jan Mayen and slightly lower than LSPE. The large deviation between the mean and the median highlights

the importance of freshly emitted particles. Indeed, as can be seen from Figure 7, higher mean than median values occurred for all the case studies considered, but the discrepancy between them was higher in the fjords most affected by anthropogenic activity. Maximum values reached in Isfjorden (up to about 4800 ng/m$^3$) were clearly the highest (excluding Tromsø), further emphasizing the presence of fresh emissions which are related to ships (intense marine traffic considered the area) and to fossil fuel combustion in Longyearbyen (mainly coal and diesel). The anthropogenic impact on Isfjorden is also confirmed through the TSP chemical composition in Longyearbyen, shown in the previous Section 3.1.

The so-called Local Settlements Pollution Effect (LSPE) is the consequence of the spread of eBC outside the anthropic fjords, detectable in the surrounding sea area. Its average eBC concentration was $70.2 \pm 1.1$ ng/m$^3$, a quite high value dominated by the effect of Isfjorden. The mean was close to the median (66.9 ng/m$^3$), and the maximum values were lower than the anthropic fjords. This evidence means that particulate matter that had undergone local transport was detected, instead of peaks of fresh emission. This aspect is further emphasized for the Long-Range Transport Events (LRTE). Moreover, in this case, mean and median values were lower ($43.3 \pm 0.8$ ng/m$^3$ and 39.8 ng/m$^3$, respectively) than LSPE, despite their emission sources being comparable to and/or greater than Longyearbyen, because these particles have been subjected to a much longer transport. The sources of LRTE were Northern Europe and northwestern Russia (Figures S1–S4). eBC concentrations related to LRTE were measured over the open sea, and the average was about four times higher than the background, thus showing a non-negligible alteration of pristine conditions. It was also higher than the anthropic fjords, except Isfjorden. However, probability of transport from mid-latitudes during the summer season is minimal (compared to winter when the Arctic haze phenomenon is present). Thus, we detected only few transport events.

Particle number concentration measurements were available only for the AREX 2019 campaign. Figure 8 shows the mean values and related 95% confidence intervals for the investigated case studies, with the exception of Jan Mayen (reached by S/Y Oceania only in 2018) and Tromsø. Figure 8a represents the total number concentration from 10 nm (obtained from the CPC), while Figure 8b,c represent the concentration of particles in the coarse (diameters larger than ~1 μm) and accumulation modes (~0.1–1 μm), respectively (obtained from the LAS channels). By subtracting the LAS from the CPC concentrations, the number concentration of particles in the Aitken regime (diameters ranging from ~10 to ~100 nm) was also determined (Figure 8d).

The coarse fraction presented very low values, because larger particles count for little in terms of number concentration: they always contributed less than 0.15% to the total particle number concentration. However, there are differences between LSPE or LRTE and the other cases: for background and anthropic fjords, the coarse particles ranged between $0.06 \pm 0.15 \times 10^{-2}$ and $0.1 \pm 0.24 \times 10^{-2}$ cm$^{-3}$; during LSPE and LRTE, however, the values nearly tripled ($0.3 \pm 0.95 \times 10^{-2}$ and $0.31 \pm 0.83 \times 10^{-1}$ cm$^{-3}$, respectively) because of the distance from the emission sources and the processes that occurred during transport.

The total number concentration was similar for the background, Hornsund fjord, and Kongsfjorden ($629.86 \pm 28.85$, $567.51 \pm 22.53$, and $521.81 \pm 25.44$ cm$^{-3}$, respectively). It was about three times higher in Isfjorden and Van Mijenfjorden ($1982.3 \pm 80.52$ and $1701.52 \pm 37.47$ cm$^{-3}$, respectively) due to the very high concentration of Aitken particles. In general, the summer period is dominated by fresh, small, and locally formed Aitken particles [10,44,98], which, in fact, constituted the largest fraction of the total in all areas/cases considered (except LRTE, Table 1); however, in Isfjorden and Van Mijenfjorden, this contribution was even greater, both in terms of absolute and percentage values (92 and 93%, respectively). LSPE presented high total number concentration values ($1080.61 \pm 158.19$ cm$^{-3}$) because of the main influence of Isfjorden, but the percentage contribution of nanoparticles (less than 100 nm, 62%) was definitely lower than Isfjorden and the other anthropic fjords, while the contribution from the accumulation mode was the highest (38%), excluding LRTE. The absolute concentration value in this dimensional range for LSPE was the greatest ($410.01 \pm 4.22$ cm$^{-3}$). This can be attributed to the

transport (not long-range) of particles outside the fjords towards the open sea in case of LSPE. In background and anthropic fjord areas, the emission of fresh nanoparticles or the formation of new particles through secondary processes dominated, making the Aitken particles by far the most important fraction. LRTE were different from the other cases: precisely because of long-range transport and consequent deposition phenomena, the total aerosol number concentration was the lowest ($245.52 \pm 4.3$ cm$^{-3}$). Moreover, it was completely attributable to the accumulation mode (almost no nanoparticles were detected). This is also in agreement with the relatively high value of the coarse fraction, mentioned above (even if it does not contribute to the total in terms of number), and with literature data showing that Arctic haze (i.e., long-range transported particles) is dominated by accumulation-mode aerosols [13].

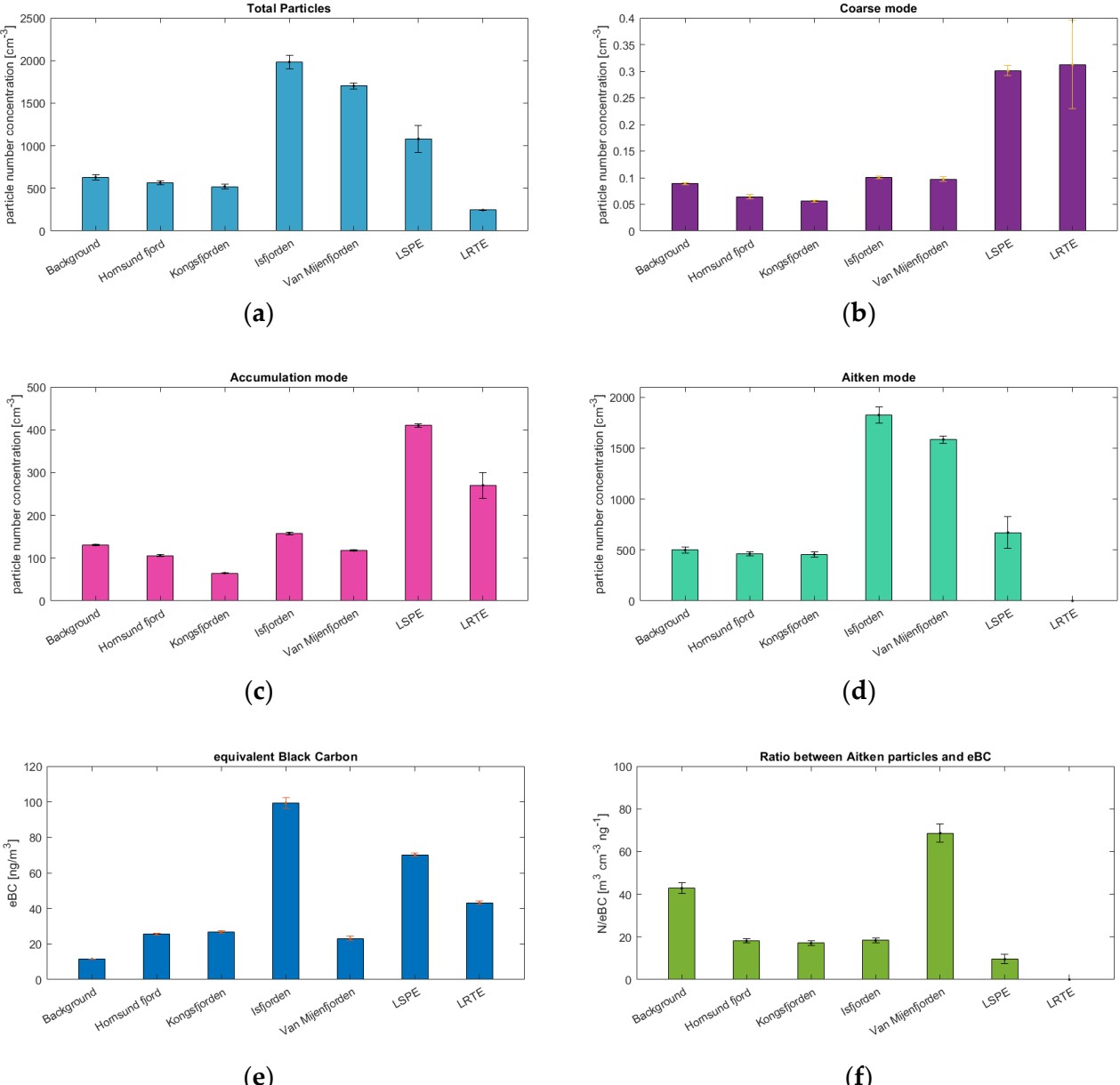

**Figure 8.** Mean values and related 95% confidence intervals of particle number concentration in the different areas/case studies considered for total particles (**a**), coarse mode (**b**), accumulation mode (**c**), and Aitken mode (**d**); mean values and related 95% confidence intervals of eBC concentration (**e**) and N/eBC ratio in the same areas/case studies (**f**).

**Table 1.** Percentages of coarse, accumulation, and Aitken fractions on total particles.

| Particles Fraction | 'Background' | 'Hornsund Fjord' | 'Kongsfjorden' | 'Isfjorden' | 'Van Mijenfjorden' | 'LSPE' | 'LRTE' |
|---|---|---|---|---|---|---|---|
| Coarse Mode | 0.01% | 0.01% | 0.01% | 0.01% | 0.01% | 0.03% | 0.13% |
| Accumulation Mode | 21% | 19% | 12% | 8% | 7% | 38% | 100% |
| Aitken Mode | 79% | 81% | 88% | 92% | 93% | 62% | 0% |

In order to estimate the anthropogenic contribution (order of magnitude) to the different areas/case studies versus other origins, the method developed by Rodríguez and Cuevas [99] and based on the $N_{Aitken}/eBC$ ratio was applied (Figure 8f). eBC is considered as a proxy of the primary aerosol in the nanoparticles range. The mean $N_{Aitken}/eBC$ for the background was $42.82 \pm 2.49$. This value is comparable to that reported by Ferrero et al. [10] (54.8) at the base of summer vertical profiles influenced by surface plumes of locally formed secondary nano-particles in Ny-Ålesund, and it clearly indicates the presence of processes of secondary aerosol formation [100]. In Hornsund fjord, Kongsfjorden, and Isfjorden, the ratios were very similar to each other ($18.09 \pm 0.95$, $17.1 \pm 1.03$, and $18.36 \pm 0.99$, respectively), despite the different concentrations of eBC, and significantly lower than the background, highlighting the influence of direct primary emissions (due to the anthropogenic impact) and the lower relevance of secondary formation in these areas compared to the open sea. In Van Mijenfjorden, an anomalous value, higher than the background, was found. There are no obvious reasons for this behavior; however, as already mentioned, S/Y Oceania sailed only half of the Van Mijenfjorden, without reaching the actual emission hotspot (Sveagruva) and measuring the related primary emissions of nanoparticles. Consequently, the relative importance of eBC compared to the total concentration of nanoparticles is lower than in other anthropic fjords.

Isfjorden and Van Mijenfjorden both showed very high values of total number aerosol concentration because they were dominated by the Aitken fraction. Nevertheless, these two elevated concentrations of nanoparticles were clearly due to different reasons, as can be seen from the $N_{Aitken}/eBC$ ratio: in Isfjorden, the Aitken mode was mainly attributable to fresh eBC emissions (lower $N_{Aitken}/eBC$), while in Van Mijenfjorden, it was mainly attributable to secondary activity (higher $N_{Aitken}/eBC$). Therefore, the $N_{Aitken}/eBC$ ratio (in addition to eBC concentrations and chemical composition) also confirmed the high anthropogenic impact in Isfjorden.

As expected, LSPE presented the lowest value (except for LRTE) of $N_{Aitken}/eBC$ ($9.55 \pm 2.26$) in accordance with its percentage of Aitken particles, which is the lowest. Obviously, the $N_{Aitken}/eBC$ ratio was 0 for LRTE because there were no nanoparticles (only particles in the accumulation and coarse mode).

### 3.2.2. Heating Rate

In this section, we present the local climate impact (direct forcing in terms of atmospheric heating) of the eBC concentrations in the areas/case studies shown in the previous paragraph.

Solar radiation is necessary to trigger the climatic effect of LAA. In the Arctic region, high winter eBC concentrations cannot interact with radiation. The local LAA radiative forcing is present during spring and summer. Its effect could be stronger in spring, when transport of particles from lower latitudes is still present and most of the surfaces are still covered by ice and snow [9,11]. In summer, however, radiation is higher and local emissions, which heat the lower atmospheric layer, causing the maximum positive forcing [9,35], are dominant. Figure 9 shows the results of the radiation measurements needed to experimentally compute the LAA HR. In this plot, Tromsø is included in the same box with the other cases. Global radiation was quite constant for all the areas/cases analyzed. Indeed, excluding Tromsø and Jan Mayen, average values ranged from $117.5 \pm 1.8 \, \text{W/m}^2$ to $160.3 \pm 8.8 \, \text{W/m}^2$. The two hotspots, which are the two southernmost localities surveyed,

presented the lowest average values measured along the cruises ($82.7 \pm 4.6$ W/m$^2$ for Tromsø and $85.5 \pm 2.5$ W/m$^2$ for Jan Mayen).

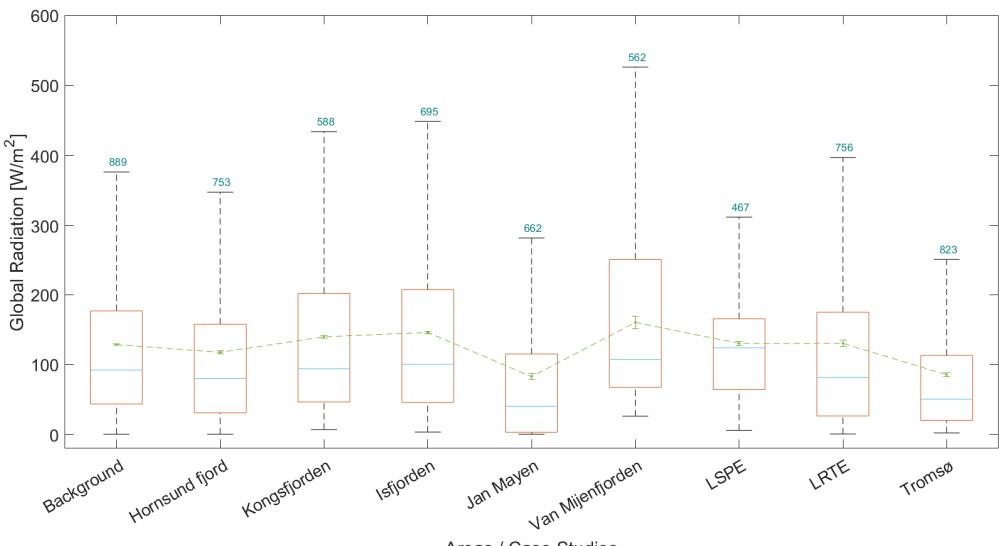

**Figure 9.** Total radiation boxplots for the areas/case-studies considered. The central mark indicates the median, while the bottom and top edges of the box indicate the 25th and 75th percentiles, respectively; the whiskers extend to the most extreme data points not considered outliers (outliers are not plotted). The mean (dashed line) and the related 95% confidence interval are also represented. The cyan numbers above the top edge of the box point out the maximum value (i.e., the maximum outlier) reached.

Thanks to the radiation (absolute values and spectral data) and absorption coefficient measurements, it was possible to determine the surface LAA HR in a completely experimental way and in any sky condition, using the method developed by Ferrero et al. [70] and described in the methodology section. It also allowed us to avoid the errors induced by the clear-sky assumption, which can be present in some radiative-transfer calculations. The last feature is crucial in the Arctic because $87.2 \pm 0.1\%$ of the measured global radiation was, on average, given by the diffuse component; in fact, the average cloudiness (in oktas) along the campaign was $7.4 \pm 0.01$. Figure 10 shows the boxplots and the average and maximum values of the computed HR (Figure S7 shows the same HR boxplots but on a logarithmic scale). Since the radiation values were quite constant, the HR variations were mainly influenced by the differences in the eBC concentrations.

Additionally, for the HR, except for LSPE, the average values were clearly greater than the median ones, in particular for Tromsø and Isfjorden, where the maximum values were reached. The HR in Tromsø was the highest, with a mean value of $68.9 \times 10^{-3} \pm 3.3 \times 10^{-3}$ K/day, followed by Isfjorden, where we found the highest values for Svalbard and Arctic Ocean areas. Here, the mean HR was $8.2 \times 10^{-3} \pm 0.3 \times 10^{-3}$ K/day, while the maximum peaks reached $5.3 \times 10^{-1}$ K/day. Although they are not high values compared to the polluted areas of the lower latitudes, the difference with respect to the pristine background of the Arctic is evident. Indeed, BG had an average HR one order of magnitude lower ($0.8 \times 10^{-3} \pm 0.9 \times 10^{-5}$ K/day) than Isfjorden and a maximum value of $0.1 \times 10^{-1}$ K/day. The HR in Jan Mayen remained higher than the anthropic fjords (excluding Isfjorden, in accordance with eBC concentrations), but decreased in proportion due to the lower radiation (compared to the higher latitudes). Among the anthropic fjords, the difference between the average HR in Kongsfjorden and Hornsund fjord increased ($2.9 \times 10^{-3} \pm 10.5 \times 10^{-5}$ K/day and $1.3 \times 10^{-3} \pm 2.8 \times 10^{-5}$ K/day, respectively) compared to the eBC mean concentrations because of differences in solar radiation. Van Mijenfjorden's mean HR ($3.2 \times 10^{-3} \pm 29.5 \times 10^{-5}$ K/day) exceeded that of Kongsfjorden and Hornsund

fjord thanks to greater radiation. The LSPE HR ($5.4 \times 10^{-3} \pm 12.7 \times 10^{-5}$ K/day) came immediately after Isfjorden because it was affected by its high eBC emissions. Additionally, LRTE can significantly alter the BG. In fact, their mean HR ($3.3 \times 10^{-3} \pm 12.1 \times 10^{-5}$ K/day) was higher than all the anthropic fjords except Isfjorden. LSPE and LRTE, although they had mean HR values greater than most of the anthropic fjords, were characterized by relatively low maximum values. These, indeed, were comparable to the maximum values reached in Van Mijenfjorden and Hornsund fjord, and they were about three to four times lower than those present in Kongsfjorden. This phenomenon is attributable to the fact that LSPE and LRTE are defined by a spread or transport from source areas; therefore, they do not show high peaks of eBC due to fresh emissions, but have more homogeneous concentrations. As already mentioned, Van Mijenfjorden is a separate case (a hybrid between an anthropic fjord and LSPE) because the ship did not reach the settlement of Sveagruva, avoiding possible direct and fresh eBC emissions from it. Finally, Kongsfjorden presented a maximum HR value ($0.8 \times 10^{-1}$ K/day), much higher than Hornsund fjord ($0.2 \times 10^{-1}$ K/day) due to the presence of Ny-Ålesund and its harbor, where ship traffic, responsible for elevated ground eBC concentrations [10], is far greater than in Hornsund, where only the Polish station is present.

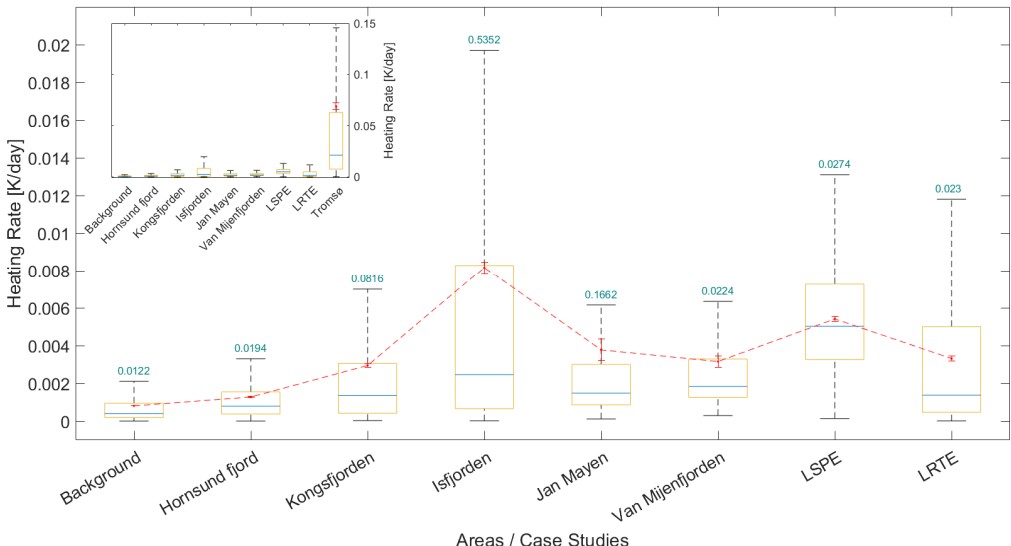

**Figure 10.** HR boxplots for the areas/case-studies considered. The central mark indicates the median, while the bottom and top edges of the box indicate the 25th and 75th percentiles, respectively; the whiskers extend to the most extreme data points not considered outliers (outliers are not plotted). The mean (dashed line) and the related 95% confidence interval are also represented. The cyan numbers above the top edge of the box point out the maximum value (i.e., the maximum outlier) reached.

Considering the maximum values (and not the averages), the HR in the hotspots or anthropic fjords of the Arctic is more similar to the HR computed at lower latitudes. For instance, the maximum HR in Isfjorden ($\approx$0.5 K/day) can be compared, although lower, to the highest values (2.1 and 1.8 K/day) found by Tripathi et al. [29] along vertical profiles obtained in the urban area of Kanpur (northern India). Therefore, peaks reached in the Arctic are not negligible. They are mainly due to fresh LAA emissions, which could rise in the future because of ice retreats and enhanced marine traffic. The role of local HR, induced by LAA, thus may become more important. Since experimental HR measurements (with our methodology) were carried out only in Milan, where LAA concentrations are higher than in the Arctic, it is useful to normalize the HR values to the unit mass of eBC for a comparison between the Arctic and mid-latitudes. Ferrero et al. [30] used the parameter HR/eBC to report the efficiency of warming per mass concentration of eBC at different cloudiness levels (oktas) in Milan. They found a

value around 0.1 K m$^3$ d$^{-1}$ µg$^{-1}$ at 7–8 oktas. We found an average cloud-cover value of 7.4 ± 0.01 oktas in the Arctic and HR/eBC values of 0.1 and 0.08 K m$^3$ d$^{-1}$ µg$^{-1}$ for maximum and average eBC concentrations, respectively, in Isfjorden. These numbers are very similar, highlighting the reliability of our data and their comparability with data from lower latitudes.

## 4. Conclusions

Particulate matter samples, light-absorbing aerosol absorption coefficients, equivalent black carbon (eBC), radiation, and particle number concentration measurements were collected during two summer Arctic campaigns (2018 and 2019), which took place in the ocean around the Svalbard Archipelago and within its anthropized fjords.

This ensemble of measurements allowed us to unravel the Arctic background level (pristine conditions over the sea) with respect to the influence of local anthropized settlements in terms of aerosol concentration, chemical composition, and direct climatic impact of light-absorbing aerosol. In addition, long range transport events (from lower latitudes) and local settlements' pollution effects (i.e., the spread of the anthropogenic impact in the area outside the fjords) were quantified too. All the real time measurements and sampling activities were carried out on the Polish research vessel S/Y Oceania as a mobile Arctic platform. Particulate matter samples were later analyzed using ion chromatography for water-soluble inorganic ions and using total carbon analysis for total carbon and organic matter content, while the LAA and radiation measurements allowed us to assess, for the first time in the Arctic, the atmospheric heating rate (HR) due to the absorption of light by the atmospheric aerosols at high temporal resolution in a completely experimental manner and in any sky condition, avoiding the inherent uncertainties in the models.

The chemical composition highlighted substantial differences between the background Arctic Ocean (e.g., eBC concentrations of 11.7 ± 0.1 ng/m$^3$ and particulate matter dominated by the sea source, with Cl$^-$ and Na$^+$ explaining 41% and 26% of the mass, respectively) and the anthropized settlements, among which Tromsø (the main city of northern Norway in the Arctic area) and Longyearbyen (the main human settlement in Svalbard Islands) had the greatest impact. In Tromsø and Longyearbyen, the relative percentages of the sea source decreased, while the anthropogenic footprint, especially in terms of eBC (21% and 14%, respectively), organic matter (most likely due to combustion origin), and nss-SO$_4$$^{2-}$ (15% and 35%), increased significantly. The differences between the two hotspots are due to their different sizes, locations, and fuels used for heat and energy production.

High-resolution measurements (eBC and particle number concentration) confirmed the local settlements' impact on the Arctic atmosphere. All the other Svalbard anthropized fjords presented roughly similar average eBC concentrations (≈25 ng/m$^3$), except Isfjorden (where Longyearbyen is located), where eBC concentrations were particularly high for the Arctic area, with an average value of 99.4 ± 3.1 ng/m$^3$. Moreover, the significant deviation between the mean and the median and the high maximum values (up to 4800 ng/m$^3$) highlighted the presence of fresh emissions which are related to ships and fossil fuel combustion. Regarding particle size, both the background and the anthropic fjords were dominated by the Aitken mode, but for different reasons: the N$_{Aitken}$/eBC ratio for the fjords was lower, emphasizing the importance of primary eBC emissions and, thus, of the anthropogenic impact; the mean N$_{Aitken}$/eBC value for the background, instead, was more than double that in the fjords and clearly indicated the presence and relevance of processes of secondary aerosol formation. Only a few Long-Range Transport Events were detected (by means of back-trajectories), in accordance with the measurement season; they were dominated by the accumulation mode due to the processes that occur during transport, and their eBC average concentration was about four times higher than the background. Thus, they show a non-negligible alteration of pristine conditions, which, however, occur only rarely in summer (when the Arctic haze is not present).

All the above data allowed the determination of climatic impact to be quantified in terms of heating rate. The HR in Tromsø was the highest ($68.9 \times 10^{-3} \pm 3.3 \times 10^{-3}$ K/day), followed by Isfjorden, where we observed the highest values for Svalbard and the Arctic Ocean areas ($8.2 \times 10^{-3} \pm 0.3 \times 10^{-3}$ K/day). Although they are not high values compared to the polluted areas of the lower latitudes, the difference with respect to the pristine background of the Arctic Ocean is evident: here the average HR is one order of magnitude lower than Isfjorden ($0.8 \times 10^{-3} \pm 0.9 \times 10^{-5}$ K/day). Moreover, considering the maximum values, the HR in the hotspots or anthropic fjords of the Arctic is more similar to the HR computed at lower latitudes. For instance, the maximum HR in Isfjorden ($\approx 0.5$ K/day) can be compared to the highest values ($\approx 2$ K/day) found in the urban area of Kanpur. The parameter HR/eBC computed in Isfjorden was similar to that in Milan (the only place where the HR was determined using the same methodology) under comparable cloud cover conditions, highlighting the reliability of our data and their comparability with data from lower latitudes.

We found that during summer, there are significant differences in light-absorbing aerosol concentrations and consequent direct climate impact (HR) between the different Arctic areas considered in this study due to the role of local sources (such as ships and human settlements on Svalbard Archipelago): they emit eBC directly within the Arctic dome, causing a strong surface warming and, thus, are necessary to explain the HR spatial variability at the local scale. Indeed, the HR peaks at the surface atmospheric layer, which are not negligible, are mainly due to fresh emissions, which could rise in the future because of ice retreats and enhanced marine traffic (tourist and commercial cruises). The role of local HR, induced by LAA, thus may become more important.

Finally, our fully experimental HR calculation methodology proved to be crucial in the Arctic area, where we found an average cloudiness of $7.4 \pm 0.01$ oktas. Hence, it could be useful for further studies in the Arctic and comparisons with results from models.

**Supplementary Materials:** The following supporting information can be downloaded at: https://www.mdpi.com/article/10.3390/atmos14121768/s1, Figure S1: Five-day air mass backtrajectories derived using the HYSPLIT for the LRTE occurred on 29 July 2018. Backtrajectories ending at 05:00 UTC 29/07/2018 at 100 (red), 500 (blue) and 1000 (green) meters above ground level. Figure S2. Five-day air mass backtrajectories derived using the HYSPLIT for the LRTE occurred on 21 June 2019. Backtrajectories ending at 09:00 UTC 21/06/2019 at 100 (red), 500 (blue) and 1000 (green) meters above ground level. Figure S3. Five-day air mass backtrajectories derived using the HYSPLIT for the LRTE occurred on 23 and 24 June 2019. Backtrajectories ending at 23:00 UTC 23/06/2019 at 100 (red), 500 (blue) and 1000 (green) meters above ground level. Figure S4. Five-day air mass backtrajectories derived using the HYSPLIT for the LRTE occurred on 19 August 2019. Backtrajectories ending at 02:00 UTC 19/08/2019 at 100 (red), 500 (blue) and 1000 (green) meters above ground level. Figure S5. Comparison between Hornsund fjord (reached by s/y Oceania in both 2018 and 2019) and Van Mijenfjorden (reached by s/y Oceania only in 2019): Hornsund, like other fjords, was covered entirely; Van Mijenfjorden, instead, is the only fjord that has been covered for half of its length, without reaching the actual settlement, Sveagruva (it is also the fjord where the ship was present for the shortest time). Figure S6. eBC boxplots for the areas/case-studies considered. The *y*-axis is in logarithmic scale. The central mark indicates the median, while the bottom and top edges of the box indicate the 25th and 75th percentiles, respectively; the whiskers extend to the most extreme data points not considered outliers (outliers are not plotted). The mean (dashed line) and the related 95% confidence interval are also represented. The cyan numbers above the top edge of the box point out the maximum value (i.e., the maximum outlier) reached. Figure S7. HR boxplots for the areas/case-studies considered. The *y*-axis is in logarithmic scale. The central mark indicates the median, while the bottom and top edges of the box indicate the 25th and 75th percentiles, respectively; the whiskers extend to the most extreme data points not considered outliers (outliers are not plotted). The mean (dashed line) and the related 95% confidence interval are also represented. The cyan numbers above the top edge of the box point out the maximum value (i.e., the maximum outlier) reached. Table S1. Number of real-time valid data (see Sections 2.3–2.5) at 1-minute time resolution for each areas/case studies. Classes containing less data are: LRTE and LSPE, because they are linked to episodic phenomena; Jan Mayen and Van Mijenfjorden, because they were

reached by s/y Oceania only in 2018 and 2019 respectively. The value is particularly low for Van Mijenfjorden, which is a special case (as already explained in the main text, Section 3): it is the fjord where the ship spent the least time and it has been covered only for half of its length (without reaching the actual settlement, Sveagruva).

**Author Contributions:** Conceptualization, N.L. and L.F.; methodology, N.L., P.M. (Piotr Markuszewski) and P.M. (Przemysław Makuch); software, N.L. and L.F.; formal analysis, N.L. and M.K.; investigation, N.L.; resources, L.F., E.B., G.M., M.R., V.D. and T.Z.; data curation, N.L.; writing—original draft preparation, N.L.; writing—review and editing, L.F., T.Z., P.M. (Piotr Markuszewski), M.R., A.G., V.D., P.P., A.M.C., I.G., A.D., S.C. and P.M. (Pietro Maroni); visualization, N.L.; supervision, L.F.; project administration, L.F.; funding acquisition, L.F. All authors have read and agreed to the published version of the manuscript.

**Funding:** This work has been founded by GEMMA Center in the framework of project MUR "Dipartimenti di Eccellenza 2023–2027". This research was also supported by Polish National Agency for Academic Exchange under the national component for the Mieczysław Bekker scholarship, 2019 edition, project id: BPN/BKK/2021/1/00004.

**Institutional Review Board Statement:** Not applicable.

**Informed Consent Statement:** Not applicable.

**Data Availability Statement:** Data are available upon request.

**Acknowledgments:** This work is an output of the GEMMA Center in the framework of project MUR TECLA "Dipartimenti di Eccellenza 2023–2027". The authors want to acknowledge Aerosol d.o.o and IOPAN for supporting the experimental campaigns. We kindly thank the crew of the r/v Oceania for all technical assistance and safety care during the cruise.

**Conflicts of Interest:** The authors declare no conflict of interest.

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
