# Peer review of "Anthropic Settlements’ Impact on the Light-Absorbing Aerosol Concentrations and Heating Rate in the Arctic"

_atmosphere, doi:10.3390/atmos14121768_

Round 1

Reviewer 1 Report

Comments and Suggestions for Authors

The authors investigated the anthropogenic influence on light-absorbing aerosols and heating rate in the Arctic utilizing measurements from s/y Oceania during the AREX campaign. This study is highly relevant to the climate science community, particularly given the background of global warming. However, there are several areas within the manuscript that require attention and resolution before the study can be considered for publication.

Line 39: Concerning the "4 times faster", it should be noted that there is variability in this value as documented in different studies. I recommend that the authors present this information as a range to reflect the variations found in the literature, rather than citing a single, definitive value.

Introduction: The central theme of the paper, which addresses the impact of anthropogenic settlements on the concentrations of light-absorbing aerosols and their associated heating rates, is not adequately reflected in the introduction. The current text predominantly discusses black carbon (BC) and does not provide a sufficient introduction to the concept of heating rates. To improve the introduction, it would be beneficial to add a summary of pertinent previous research.

Line 106 and 107: What are “models”, and what are “other models”?

Materials and Methods: How many valid data (both online and offline) are used in this study? The authors classified their measurements into 5 clusters, how much data are there in each cluster? For a thorough understanding and transparency, it would be beneficial for the authors to provide the total count of valid data points used and a breakdown of the number of data points within each of the five clusters.

In Line 448, the authors have chosen to highlight the Arctic Ocean, Tromsø, and Longyearbyen when discussing variations in chemical composition. Given that the study categorizes the measurements into five distinct clusters, it would be anticipated that the differences among all these clusters would be presented for a comprehensive comparison. The rationale for focusing solely on the Arctic Ocean, Tromsø, and Longyearbyen is not immediately apparent, and the argument for identifying Tromsø and Longyearbyen as the primary emission hotspots requires further substantiation to be convincing. It would be beneficial for the readers if the authors could clarify the selection criteria for these locations and provide additional evidence to support the claim regarding their significance as emissive hotspots.

Line 817: It may be beneficial to consider using a logarithmic scale for the y-axis in Figure 10 to potentially enhance the data representation.

Regarding the back trajectory analysis, it would be informative to include the altitude profile for each trajectory.

Comment on Figures: The font size of the labels, legends, and numerical data within the figures is currently too small to be easily read. Additionally, the figure captions require more explicit descriptions. Take Fig. 7 as an example; the significance of the cyan numbers above each box is not clear. While the text of the paper specifies that these represent maximum values, such details should be directly stated within the figure captions for clarity and to facilitate better understanding without needing to refer back to the main text.

Comments on the Quality of English Language

"Anthropogenic" should be used when referring to effects, changes, or phenomena that are the result of human activity, particularly in the context of environmental impact. In contrast, "anthropic" broadly relates to human existence or human-related factors and does not necessarily imply causation by human activity. The manuscript appears to contain instances where the terms "anthropic" and "anthropogenic" are used incorrectly and inconsistently, e.g. line 611, line 2 and 27. We suggest that the authors conduct a comprehensive review of the document to ensure that these terms are applied correctly throughout.

Reviewer 2 Report

Comments and Suggestions for Authors

This manuscript describes the impact of anthropogenic emissions on heating rates in the Arctic using a recently developed method based on calculations which relate solar radiation measurements, black carbon concentration and absorption coefficient determination, aerosol fine particle concentration and composition, size distribution and composition to heating rate. Black carbon was the considered to be the significant contributor to heating rate. While certain assumptions were made in the calculations, the report is an experimental approach rather than a modeling approach. While there appears to be only one similar study reported (also by a co-author of this manuscript) the results of the studies are consistent and credible, lending some confidence to this report. Importantly, the measured inputs into the calculations reported here were compared with those in numerous independent were found to be consistent. Thus, the basic inputs into the calculations presented in this manuscript are supported. The authors conducted a campaign during consecutive years from a research vessel that surveyed a range of locations, including pristine ocean expanses, populated fjords and the environs of Tromso, a major urban center and the Svalbard Archipelago, which is strongly influenced by settlements and marine traffic. The results of the analysis was organized according to the following categories: arctic background, populated fjords, areas surrounding populated centers located within the fjords (“hotspots”) and periods characterized as long range transport events, during which prevailing winds transported pollution from densely populated sites in lower latitudes. Within each of the location categories, marker aerosol components were used to resolve background contribution to the heating rate from anthropogenic contributions. HYSPLIT trajectories were used to identify long distance transport events. While this reviewer is not an expert at radiation measurements or aerosol field measurements, experimental procedures appear to be meticulous, and very unlikely to have introduced significant systematic errors. The study demonstrates that anthropogenic emissions within the arctic region causes significant increases in the heating rate, with the more densely populated centers making the largest contributions. Influences from marine traffic, long range transport events, though smaller, were nonetheless measurable. The results have been interpreted to predict that anticipated development within the arctic region combined with climate change will accelerate the heating rate.

The study reported in this manuscript represents an original approach to assessing the sources of warming in the Arctic and can serve as a benchmark for evaluating model outputs or further refining of calculated warming effects based on environmental measurement. Publication is recommended.

Before publication, a English usage needs to be addressed. In a number of cases, problems with English usage have made the text confusing or difficult to read. The manuscript indicating some, but by no means all, instances of incorrect usage has been attached.

Some suggestions may be applied globally:

The preferred American spelling is “sulfate” rather than “sulphate”. The authors should consider making this change.

The authors appear to use “anthropic” and “anthropogenic” interchangeably. It would be preferable to use “anthropogenic” when referring to physical effects or airborne contaminants that originate from human activity and “anthropic” to describe locations or areas that are populated.

Line 87: It would be helpful to indicate in legible text the location of major settlements visited on the campaigns.

Line 165: The citation for pretreatment of filter blanks [27] describes the pretreatment of Teflon rather than quartz fiber filters.

Line 251: humidity à relative humidity?

Lines 322 – 323: The writing here is confusing. The reviewer assumes that the advantages of other approaches are not the measurements that follow the parenthetical phrase.

Lines 821 – 823: Why was the mean heating rate in Tromso compared to the average heating rate values for Svalbard rather than comparing the mean values for both areas?

Comments on the Quality of English Language

See pdf in Comments and suggestions for authors.

Reviewer 3 Report

Comments and Suggestions for Authors

The manuscript is sound and detailed. Background, data collection and analysis are described in detail. Overall interesting findings. I do not have major comments. 

Comments on the Quality of English Language

The manuscript is clear and overall written in good English. There are however several sentences that need some editing and improvements in the use of the English.
